# Gold and Silver Nanoparticles as Biosensors: Characterization of Surface and Changes in the Adsorption of Leucine Dipeptide under the Influence of Substituent Changes

**DOI:** 10.3390/ijms25073720

**Published:** 2024-03-27

**Authors:** Edyta Proniewicz

**Affiliations:** Faculty of Foundry Engineering, AGH University of Krakow, 30-059 Krakow, Poland; proniewi@agh.edu.pl

**Keywords:** biosensors, gold nanoparticles, AuNPs, silver nanoparticles, AgNPs, surface analysis, phosphonate dipeptides of leucine, adsorption, surface-enhanced Raman spectroscopy, SERS

## Abstract

Early detection of diseases can increase the chances of successful treatment and survival. Therefore, it is necessary to develop a method for detecting or sensing biomolecules that cause trouble in living organisms. Disease sensors should possess specific properties, such as selectivity, reproducibility, stability, sensitivity, and morphology, for their routine application in medical diagnosis and treatment. This work focuses on biosensors in the form of surface-functionalized gold (AuNPs) and silver nanoparticles (AgNPs) prepared using a less-time-consuming, inexpensive, and efficient synthesis route. This allows for the production of highly pure and stable (non-aggregating without stabilizers) nanoparticles with a well-defined spherical shape, a desired diameter, and a monodisperse distribution in an aqueous environment, as confirmed by transmission electron microscopy with energy-dispersive X-ray spectroscopy (TEM-EDS), X-ray diffraction (XRD), photoelectron spectroscopy (XPS), ultraviolet-visible (UV-VIS) spectroscopy, and dynamic light scattering (DLS). Thus, these nanoparticles can be used routinely as biomarker sensors and drug-delivery platforms for precision medicine treatment. The NPs’ surface was coated with phosphonate dipeptides of L-leucine (Leu; l-Leu–C(R^1^)(R^2^)PO_3_H_2_), and their adsorption was monitored using SERS. Reproducible spectra were analyzed to determine the orientation of the dipeptides (coating layers) on the nanoparticles’ surface. The appropriate R^2^ side chain of the dipeptide can be selected to control the arrangement of these dipeptides. This allows for the proper formation of a layer covering the nanoparticles while also simultaneously interacting with the surrounding biological environment, such as cells, tissues, and biological fluids.

## 1. Introduction

Ref. [1] defines a nanoparticle (NP) as a material containing particles in an unbound state, aggregate, or agglomerate. These particles can be naturally occurring, accidental, or manufactured. The number size distribution of this material contains 50% or more particles with one or more external dimensions in the size range of 1–100 nm. Nanoparticles can be divided into different types: carbon-based, organic-based, inorganic-based, composite-based, and metal-based nanoparticles, as well as dendrimer nanoparticles with advanced adhesion [2]. Metal and metal oxide nanoparticles are among the best-studied types of nanoparticles and have attracted considerable attention in various fields such as materials science, physics, electronics, organic and inorganic chemistry, chemical catalysis, agriculture, food industry, molecular biology, medicine, and pharmaceuticals [3,4,5,6,7,8,9,10]. For example, the use of precious metals such as (Ag) and gold (Au) for their healing properties was already an issue in ancient Rome, Greece, Egypt, and China. These metals were applied directly to wounds to improve healing and to vases to disinfect water and preserve food [11,12]. The use of Ag salts, especially nitrate salts, for medicinal purposes has been documented since the first century [13]. In the early 20th century, colloidal silver was first used as an external and internal antimicrobial agent for infection control, disinfection of hospital facilities, and treatment of diseases such as tuberculosis and gonorrhea [13,14,15,16]. Studies have also demonstrated the efficacy of silver nanoparticles (AgNPs) and gold nanoparticles (AuNPs) against leprosy bacteria [17]. AgNPs are also effective against other bacteria, fungi, and viruses [18,19,20], similar to AuNPs, which have a broad spectrum of activity against microorganisms and viruses [21,22,23]. Silver and gold NPs have a wide range of applications, including coatings for implants, central venous catheters, surgical mesh, glass surfaces, fabrics, textiles, food storage devices, and cosmetics [22,24,25,26,27,28].

However, when considering the use of nanoparticles in biomedicine, it is important to approach the topic objectively and weigh the potential risks and benefits. Research has shown that the size of NPs and their surface-to-volume ratio, morphology, dosage, and accumulation time in the body are the most important factors to consider for their safe use. Small NPs have demonstrated potent antimicrobial activity due to their ability to reach the bacterial nucleus more easily and offer a larger surface area, resulting in stronger bactericidal properties [29]. NPs larger than 10 nm tend to accumulate on the cell surface and reduce their permeability. However, nanoparticles smaller than 10 nm penetrate the interior of the bacteria and affect DNA and enzymes, leading to apoptosis [30]. Mammalian cells take up NPs with sizes ranging from 10 to 500 nm [31]. Smaller NPs have a higher toxicity potential compared to the larger ones.

As NP size decreases, their surface area increases significantly. Studies have shown that NPs with larger surface areas interact with nearby particles and cause higher cytotoxicity. NPs with identical surface areas but varying shapes may exhibit different activities due to differences in their effective surface areas and the number of active faces. For instance, Gratton and co-workers demonstrated that the uptake potential decreases for small Ag nanostructures in the following order: rods, spheres, cylinders, and cubes [32]. In other studies, it has been found that AuNPs of less than 2 nm can trigger necrosis and mitochondrial damage and induce oxidative stress, while AuNPs of 8–37 nm have been linked to damage in the liver, spleen, and lungs. On the other hand, AuNPs of 3–5 nm and 50–100 nm are non-toxic [33]. Furthermore, AuNPs of 14, 50, and 74 nm are affected by nonspecific adsorption of serum proteins, which can impact their activity [34]. Several clinical studies have also shown that various Au nanostructures have good safety profiles with minimal adverse effects. These effects were either transient, clinically manageable, or considered unrelated to treatment with AuNPs. The activity of NPs is affected by the surface charge, which is dependent on the synthesis method and added stabilizers [35,36,37]. For instance, research has shown that positively charged AgNPs exhibit higher bactericidal activity than negatively charged ones [35]. However, the issue of dose dependency persists as it is influenced by the size, morphology, shape, and surface charge of NPs [38].

Functional groups immobilized on the surface of Ag and Au nanoparticles may also influence their cytotoxic properties. Therefore, it is necessary to conduct individual studies to understand the pharmacokinetics and toxicological profile of each system. For instance, a study showed that 27 nm AuNPs functionalized with recombinant human tumor necrosis factor-alpha (rhTNF) and thiolated polyethylene glycol can safely deliver AuNPs systemically without dose-limiting toxicity [39]. The safety profiles of AgNPs and AuNPs functionalized with cysteine and glutathione were evaluated on mammalian cells. The results showed that AgNPs prepared in the presence of GSH exhibited increased toxicity, while AuNPs exhibited reduced toxicity [40].

While various metal-based nanoparticles have been investigated and used for medical purposes [41,42,43,44,45,46,47,48,49,50,51], this article focuses on AgNPs and AuNPs because of their unique physicochemical properties. These nanoparticles enable the rapid detection of biomarkers at very low detection limits by optical absorption, fluorescence, and surface-enhanced Raman scattering (SERS) [52,53,54]. A biomarker is an indicator of a physiological, pathological, or pharmacological response to a therapeutic intervention [55]. Examples of classical biomarkers are blood lactate levels, non-invasive blood glucose levels, pathogens in the blood, and specific molecular changes in cells at the level of DNA, RNA, metabolites, or pro-proteins [55,56,57,58,59]. Biomarkers can diagnose and monitor various diseases (including cancer, metabolic disorders, infections, and central nervous system disorders) during and after treatment [60,61,62]. Certain biomarkers are used in molecular diagnostic tests [63].

From a molecular diagnostic perspective, NPs are used primarily because of their unique response to a biological target, low toxicity, and relatively large surface area relative to volume when interacting with the target. To interact with specific tissue sites, NPs require modification [64]. This is typically achieved through the immobilization of saccharides, amino acids, peptides, oligonucleotides, antibodies, proteins, and nucleic acids [65]. These biomolecules enhance the stability, biocompatibility, and targeting of NPs, while also protecting against undesirable effects [66].

This study focuses on surface-functionalized AgNPs and AuNPs obtained by a synthetic route that ensures high purity, narrow size distribution, surfactant/ion-free aqueous solution, and reproducibility of the result. This is crucial for routine medical applications. The surface of these NPs was coated with a monolayer of L-leucine phosphonate dipeptides (Leu, (S)-2-amino-4-methylpentanoic acid). These dipeptides consists of [2-(2-amino-4-methylpentanamido)ethyl]phosphonic acid (Leu1), [2-(2-amino-4-methylpentanamido)propan-2-yl]phosphonic acid (Leu2), [2-(2-amino-4-methylpentanamido)butan-2-yl]phosphonic acid (Leu3), [1-(2-amino-4-methylpentanamido)propyl]phosphonic acid (Leu4), [(1S)-1-(2-amino-4-methylpentanamido)propyl]phosphonic acid (Leu5), 1-(2-amino-4-methylpentanamido)-2-hydroxyethyl]phosphonic acid (Leu6), [1-(2-amino-4-methylpentanamido)-2,2-dimethylpropyl]phosphonic acid (Leu7), [(2-amino-4-methylpentanamido)(cyclopropyl)methyl]phosphonic acid (Leu8), 4-(2-amino-4-methylpentanamido)-4-phosphonobutanoic acid (Leu9), and [(2-amino-4-methylpentanamido)(adamantane)methyl]phosphonic acid (Leu10) (see Figure 1 for the structures).

The phosphonate acids of leucine (Leu) contain heteroatomic donor groups, specifically *C*-terminal –PO_3_H_2_ and *N*-terminal –NH_2_. Aminophosphonate compounds have the potential to exhibit antibacterial, antiviral, anticancer, and antibiotic effects, as well as inhibit the active center of enzymes and bone resorption. This is due to their ability to form electrostatic interactions and H-bonds and to form complexes with various metal ions. The effects are also contributed to by the tetrahedral geometry of the –PO_3_H_2_ group [67,68,69]. Phosphonate acids of leucine have potential applications in the treatment of various diseases associated with slow bone loss [69]. Leucine plays a significant role in the potential use of these compounds as drugs, as it promotes the growth and healing of skin, muscles, and bones [70,71] and prevents the breakdown of muscle proteins that occur during severe stress or trauma. Leucine can also reduce inflammation caused by autoimmune and heart diseases, cancer, and diabetes [72,73]. It regulates autophagy, stimulates protein synthesis, and promotes cell growth in mammals by activating the mechanistic target of rapamycin complex 1 (mTORC1) [74]. The ability of leucine to bind to a highly conserved stress-responsive protein (Sestrin2) is crucial for the activation of mTORC1 in cells. Therefore, Leu can serve as a Sestrin2 sensor for the mTORC1 pathway [75].

The surface-enhanced Raman scattering (SERS) technique was employed to investigate surface-functionalized silver and gold nanoparticles. This technique is highly effective in detecting cells, tissues, and biological fluid components with concentrations as low as 10^−14^ M. It also aids in the understanding and extension of knowledge on adhesion and adsorption occurring at the solid–liquid and solid–gas interfaces for these compounds, which is a major challenge in biomaterials science [76,77]. Accurate characterization of adsorption is crucial. Changes in the SERS intensity of an adsorbate can be misinterpreted when analyzing its concentration alone. It is important to consider the organization, arrangement, or geometry of the adsorbate on the metal surface when interpreting changes in SERS intensity. Neglecting this aspect can lead to errors and diminish the biological and medical significance of surface-modified nanoparticles.

In this study, surface analyses were performed on carrier-less AgNPs and AuNPs to characterize their surface morphology, chemical composition, and spatial structure [78]. The surface of the NPs was imaged and analyzed by transmission electron microscopy with energy-dispersive X-ray spectroscopy (TEM-EDS). The structure of the NPs was determined by X-ray diffraction (XRD) analysis. For colloidal NPs, dynamic light scattering (DLS) and ultraviolet-visible (UV-VIS) spectroscopy were also used to determine the size distribution. In addition, photoelectron spectroscopy (XPS) was used to identify minor surface contamination.

## 2. Results

### 2.1. Surface Analysis of NPs

Figure 2 and Figure 3 show the spectroscopic properties of the colloidal silver (AgNPs) and gold (AuNPs) nanoparticles. The spectroscopies used were X-ray diffraction (XRD) (inset A), energy-dispersive X-ray (EDS) (inset B), and X-ray photoelectron spectroscopy (XPS) (inset C). Insets D–F show transmission electron microscopy (TEM) images, particle size distribution (DLS) analyses, and ultra-visible (UV-VIS) spectra.

Figure 2A shows the XRD pattern with four diffractions corresponding to the [111] (at 38.1°), [200] (at 44.3°), [220] (at 64.5°), and [311] (at 77.7°) face-centered cubic (FCC) crystalline hkl planes [79]. The average crystallite size of these nanoparticles is 19.8 nm, as calculated using the Scherrer equation. Figure 2B illustrates the purity of the AgNPs measured by EDS analysis. It shows a characteristic high-intensity absorption peak at 3.0 keV for nanocrystalline Ag [80]. The XPS spectrum of the AgNPs (see Figure 2C) shows that only silver atoms are present. The binding energies for Ag^0^ 3d 5/2 and Ag^0^ 3d 3/2 agree with the corresponding core levels of Ag^0^ solid crystals (368 and 374 eV) [81] and are at 368.3 eV and 374.1 eV, respectively. The narrow width of the XPS peaks further confirms the presence of only a single element of silver in the system. In addition, the TEM image of the AgNPs (Figure 2D) shows that they are spherical with smooth surfaces and have an average size of 19–21 nm, which is consistent with the corresponding XRD-derived particle size. DLS analysis (Figure 2E) shows a narrow size distribution of AuNPs with an average hydrodynamic diameter of approximately 20 nm, confirming the TEM result. The nanoparticles exhibited a surface plasmon band at 404 nm (see Figure 2F), which is consistent with previous studies showing that AgNPs with a size of 15–20 nm absorb in the range of 401–406 nm [82]. The interaction between the dipeptide and AgNPs caused aggregation by decreasing the distance between the particles, resulting in a shift and broadening of the AgNP plasmon resonance to longer wavelengths (565 nm) [83]. The UV-Vis spectra of phosphonate dipeptides of Leu do not differ because they are characteristic of the colloid (uncharged amino acid solutions of Ala, Asn, Ile, Leu, Met, Pro, Ser, Thr, and Val show negligible absorption in this region) [84], so an example spectrum for Leu1-AgNPs system is provided in Figure 2F.

The absorption of localized surface plasmon resonance (LSPR) varies depending on the chemical element [85]. For example, the extinction spectrum of the spherical AuNPs prepared in this study with an average diameter of 20 nm (see Figure 3D) shows a maximum at 526 nm (Figure 3F). This result is consistent with other studies showing strong absorption in the 520–530 nm range of AuNPs with sizes of 12–41 nm [86,87]. The addition of a dipeptide to colloidal AuNPs leads to aggregation, which causes a color change from yellow to red (with absorbance at 620 nm). This facilitates the detection of various analytes [88]. The DLS analysis (Figure 3E) and the XRD pattern (Figure 3A) confirm the small size of the AuNPs and their narrow size distribution (15–20 nm). In addition, the XPS spectrum (Figure 3B) and XRD pattern indicate a pure cubic close-packed structure. The peaks at 2.1 keV (Au M) and 9.7 keV (Au L_α_) [89] and the diffraction at 2Θ 38.1° [111], 44.3° [200], 64.5° [220], and 77.7° [311] [90] were used as the basis for this study. The XPS spectrum of the valence band for AuNPs, as shown in Figure 3C, shows peaks at 84.2 eV (Au^0^ 3d_5/2_) and 87.8 eV (Au^0^ 3d_7/2_), which are assigned to Au^0^ [90]. This confirms that all Au^3+^ ions were reduced to the metallic form (Au^0^), which is the critical step in the synthesis of AuNPs.

### 2.2. SERS Studies

Figure 4 and Figure 5 show the SERS spectra of the Leu1–Leu10 phosphonate dipeptides. The dipeptides are labeled as l-Leu–C(R^1^)(R^2^)PO_3_H_2_, where R^1^ is –H for Leu1 and Leu4–Leu10 and –CH_3_ for Leu2 and Leu3; R^2^ comprises aliphatic groups such as –CH_3_ (analog of alanine for Leu1 and Leu2), –C_2_H_5_ (analog of aminobutyric acid for Leu3–Leu5), –CH_2_OH (analog of serine for Leu6), –C_2_H_4_COOH (analog of glutamic acid for Leu9), –C(CH_3_)_3_ (analog of isoleucine for Leu7), cyclopropane, and adamantane. Their schematic structures are shown in Figure 1. The dipeptides were immobilized on the surface of AuNPs and AgNPs in an aqueous solution at pH~7. At pH 7, the dipeptides form a cationic species, in which the N- and P-terminal groups are protonated (–NH_2_ and –PO_3_H^−^). The observed wavenumbers and proposed SERS band assignments are summarized in Table 1 and Table 2.

Previous works have shown the behavior of phosphonate dipeptides of L-glycine (Gly, where –H is the side chain of L-glycine), isomers of alanine (Ala, –CH_3_), and L-valine (Val, –CH(CH_3_)_2_) immobilized on the surface of 10 nm AgNPs using the SERS technique [91,92,93,94]. These studies have demonstrated a characteristic mode of adsorption of these peptides on the surface of AgNPs. Research has demonstrated that Gly phosphonate dipeptides interact with the silver surface primarily through the lone electron pairs on the oxygen of the P=O bond and the nitrogen of the amino group. Similarly, Ala phosphonate dipeptides adsorb to the surface mainly due to the free electron pair on P=O. Val phosphonate dipeptides, which have a more extended side chain than Gly and Ala, make contact with the AgNPs’ surface through the P–O and amide bonds. This research presents the adsorption of phosphonate dipeptides of Leu on AgNPs and AuNPs, whose side chain is more extended. The SERS bands were assigned to the normal vibrations based on previously reported SERS assignments for L-leucine and its dipeptides: L-leucine–L-leucine and L-leucine–L-methionine, as well as the phosphonate dipeptides of L-glycine (Gly), L-alanine (Ala), and L-valine (Val) [92,93,94,95,96].

In this section, the SERS bands of the phosphonate dipeptides are briefly discussed. Only bands that are unique to the different phosphopeptide fragments (R^1^ and R^2^) and provide specific structural information are considered, as well as those that show a change in intensity compared with the FT-Raman spectra (as shown in Figure 6). This approach allows the determination of the adsorption of these compounds.

#### 2.2.1. Adsorption on the Surface of AgNPs

Leu1 is a phosphonate dipeptide with a simple structure, where R^1^ is –H and R^2^ is –CH_3_. The SERS spectrum of this dipeptide on AgNPs is dominated by two bands at 740 and 1285 cm^−1^, corresponding to Amide V [out-of-plane N–H bending, ρ_oopb_(NH)] and C_a_–N_a_ stretching vibrations [ν(C_a_N_a_)], respectively (Figure 4, upper trace) (see Table 1 for band assignments). Both bands are shifted down by 30 and 10 cm^−1^ in the wavenumber (Δ_ν_) and broadened by 20 and 8 cm^−1^ in full width at half band maximum (Δ_fwhm_), respectively, compared to their FT-Raman spectra (see Figure 6). This indicates that free electron pairs at the nitrogen and oxygen atoms of the amide bond interact with the AgNP surface. Considering the hybridization of sp^2^ orbitals for oxygen and sp^3^ orbitals for nitrogen and the significant intensity of the 740 cm^−1^ band in the SERS spectrum on AgNPs, the lone electron pair of nitrogen is expected to interact directly with the Ag surface (sp^3^ orbital perpendicular to the AgNP surface), and the sp^2^ orbital with the lone electron pair of oxygen is deflected about 10 degrees from the normal surface of the AgNP. Additional bands confirming these interactions were observed at 1671 cm^−1^ [Amide I], 1580 cm^−1^ [Amide II], and 572 cm^−1^ [Amide VI] [97]. There is also a visible shoulder on the lower wavenumber side of the 1285 cm^−1^ band (at 1260 cm^−1^) associated with the higher wavenumber ν(P=O) vibrations. Other bands of the phosphate group were observed at 1168 [ν(P=O)], 1026 [ρ_b_(POH], 979 [δ(CPO_3_H^−^), δ_oop_(C_α_NH_2_), ν(C_α_N)], 928 [ν(P–O) + ρ_b_(POH)], and 894 cm^−1^ [δ_oop_(C_α_NH_2_), ρ_t_(CPO_3_H^−^), ρ_t_(C_α_C_a_(=O)N_a_)]. The vibrations of the amino group contribute to some of these bands, as can be seen from the assignment above. The enhancement of the band at 1642 cm^−1^ confirms the involvement of the –NH_2_ group in the interaction with the AgNP surface (see Table 1 for the assignment of the band). It can be concluded that the PO_3_H^−^ and NH_2_ groups are located near the AgNP surface.

The Leu2 structure differs from Leu1 by adding a –CH_3_ group at the C_P_ carbon atom (carbon adjacent to the phosphonate group). This addition causes a change in the intensity of the bands observed in the SERS spectrum of this dipeptide on AgNPs. The most intense bands were observed at 686 [Amide IV], 1221 [Amide III], and 1675 cm^−1^, indicating an interaction between the amide bond and AgNPs. Based on the hybridization, both the C_a_=O and C_a_–N_a_ bonds are tilted by about 60 degrees to the AgNP surface normal. Medium intensity SERS signals were observed at 570, 851 [ρ_r_(C_α_NH_2_), ρ_t_(C–P=O), ν(CN)], 896, 928, 973, 1289, 1482 [δ/ρ_r_(CH_3_)], and 1566 cm^−1^, suggesting that the amino and phosphonate groups are located at some distance from the AgNPs.

Leu3 and Leu4 have an elongated R_2_ side chain with an additional carbon atom (CH_3_ → C_2_H_5_) compared with Leu2 and Leu1, respectively. Leu4 (R) and Leu5 (S) differ in their absolute configuration. The elongation of the R^2^ chain in the case of Leu3 leads to a strong enhancement of the band at 1141 cm^−1^ [ρ_r_(C_α_NH_2_), ν(CC), ν(CN), ρ_r_(C_α_(H,N_a_)C_a_)], which is accompanied by medium intensity bands at 562, 692, and 952 [δ_oop_(C_α_NH_2_)] and 1172, 1269, 1580, and 1632 cm^−1^. It is hypothesized that the interaction between Leu3 and the AgNP surface is due to the C_α_(NH_2_)–C_a_(=O)N_a_H fragment. The P=O group, which is located near the surface, only supports this interaction, as shown by the low intensity of the bands at 1269 and 1172 cm^−1^. In Leu4, the elongation of the R^2^ chain leads to a clear enhancement of the bands at 587, 983, 1161, and 1253 cm^−1^. The dipeptide interacts directly with the surface of AgNP via the free electron pair on the oxygen atoms of the P=O and C_a_=O groups. The P–OH bond and the amino group are located near this surface (at 338–350 cm^−1^ band of Ag–O [98]).

Only minor differences were observed between the SERS spectra of Leu4 and Leu5 adsorbed on AgNPs. In particular, the band at 584 cm^−1^ showed a decrease in intensity of approximately 30% for Leu5, and the band at 1029 cm^−1^ also experienced a decrease in intensity. In contrast, the band at 771 cm^−1^ was more pronounced for Leu5 than for Leu4. In addition, a band at 815 cm^−1^ appeared, which was due to the vibrations of the P–O bond. These changes in the SERS spectral pattern can be attributed to differences in adsorption. The P=O group interacts directly with the surface of the AgNP, whereas the C_a_=O bond is slightly tilted from its perpendicular orientation. In addition, the deprotonated P–O^−^ group supports the interaction with the AgNP surface.

The SERS spectrum of Leu6 (a hydroxyl group in R^2^ (–CH_2_OH)) on AgNPs shows intense bands at 578, 741, 982, 1000, 1108, 1290, and 1666 cm^−1^, and less intense bands at 1191, 1342, 1432, 1481, and 1610 cm^−1^. Two spectral features at 930 and 1000 [ρ_b_(POH), ν(PO)] indicate contact between P–O and POH moieties and AgNPs. In addition, four bands at 578, 741, 1108 [ν(N_a_C), ρ_r_(CC(Y)C), δ(N_a_C(P)C), ν(CC), ρ_r_(CCH_3_)], 1290, and 1666 cm^−1^ indicate the interaction of the amide bond with the AgNP surface. The remaining SERS signals at 1290 cm^−1^ [ν(CO)+ρ_b_(COH)] and 1432 cm^−1^ [ρ_w_/ρ_t_(C(H_2_)O) + ρ_b_(COH)] and 1610 cm^−1^ [δ(NH_2_) + ρ_b_(C–NH_2_)] are enhanced and are due to the vibrations of the C–OH and NH_2_ groups (Table 1), respectively. These groups are located near the AgNP surface.

Leu7 contains a spatially extended hydrophobic R_2_ fragment (C(CH_3_)_3_) with a low affinity for Ag. This means that the R^2^ fragment should face away from the AgNP surface so that the NC_α_Ca(=O)NCP backbone lies flat at this surface, which should be reflected in the enhancement of the bands of this fragment. Indeed, intense bands due to vibrations of the amide bond (at 584, 745, and 1224 cm^−1^), the P=O bond (at 1256 cm^−1^), and the CNH_2_ group (at 963 cm^−1^) are observed in the SERS spectrum of Leu7.

In the SERS spectrum of Leu8, as in the SERS spectrum of Leu2, the Amide III band (at 1226 cm^−1^) is the most intense, followed by the Amide IV mode at 560 cm^−1^. This means that the lone pair of electrons on the nitrogen is in direct contact with the surface of the AgNP, i.e., the angle between the N_a_–H and C_a_–N_a_ bonds, and this surface is about 19.5^o^, while the C_a_=O bond forms an angle of 40.5^o^ with this surface. In contrast, the amino and phosphonate groups are located close to the AgNP surface, which is reflected in the medium-strength SERS signals of these groups (NH_2_: 971, 1069, and 1632 cm^−1^ and PO_3_H: at 797, 971, 1147, and 1270 cm^−1^).

In the Leu9 structure, an additional carboxyl group is introduced into the R^2^ fragment, compared with the Leu4 dipeptide structure. With the appearance of this group, vibrational bands of this group appear in the SERS spectrum of Leu9, i.e., at 1725, 1447, and 910 cm^−1^, indicating that the R^2^ fragment of Leu9 is in contact with the surface of AgNP. This is because the first of the above bands is due to the ν(C=O) vibrations of the carboxyl group. The band at 1447 cm^−1^ is assigned to the ρ_w_/ρ_t_(CCO) + ρ_b_(COH) mode. The last of these bands, at 910 cm^−1^, is assigned to ν(C–COOH). In the SERS spectrum of Leu9 on AgNPs, the bands due to amide vibrations, e.g., at 571, 738, 1218, and 1550 cm^−1^, the bands originating from the amide bond, e.g., at 641 [(N_a_C(P)C), δ_oop_(C_a_N_a_(H)C), ν(CP)] and 544 cm^−1^, and the phosphate group, e.g., at 1268, 1161, and 1054 cm^−1^, are also enhanced.

Compared with Leu7, the R^2^ fragment in the Leu10 dipeptide is an even more spatially complex group (adamantane). In the SERS spectrum of this peptide, the vibrational bands of the C(NH_2_)C_a_(=O)N_a_(H) fragment (at 573, 738, 980, 1080, 1244, 1302, and 1347 cm^−1^ (see Table 1 for band assignments)) and the POH moiety (at 780, 810, and 1000 cm^−1^) are enhanced, indicating that only these dipeptide fragments interact with the AgNP surface. Surprisingly, only the Leu10 dipeptide has no bands in its spectrum due to the vibrations of the P=O bond, suggesting that this bond is distant from the AgNP surface.

#### 2.2.2. Adsorption at the Surface on AuNPs

Figure 5 presents the SERS spectra of the Leu1–Leu10 phosphonate dipeptides adsorbed on the surface of 20 nm AuNPs, which corresponds to the size of AgNPs. The SERS spectra show clear enhancement of the bands at 1254 and 1172 cm^−1^ and 1265 and 1176 cm^−1^ due to ν(P=O) only for Leu1 and Leu9 on AuNPs. These spectra show two additional medium-intensity bands related to the vibrations of the phosphonate group, specifically at 1000 and 1041 cm^−1^. These bands correspond to the ν(PO) mode, with a contribution of ρ_b_(POH), and ρ_b_(POH) modes. Similar bands appear in the spectra of the other dipeptides at comparable wavenumbers (see Table 2) but with varying intensities. In the Leu2 SERS spectrum, the 1020 cm^−1^ band has medium intensity, whereas the 995 cm^−1^ band is weaker. The spectral feature at 998 cm^−1^ has a low intensity in the Leu3 SERS spectrum, which is dominated by the band at 1026 cm^−1^. In the SERS spectra of Leu8 and Leu9 on AuNPs, the intensity of the band at the higher wavenumber is stronger than that of the band at lower wavenumber. The relative intensities decrease in the following order: Leu3 > Leu2 > Leu9 > Leu8. However, for Leu5, both bands have the similar intensity. For Leu4, Leu6, Leu7, and Leu10, the band at the lower wavenumber is more intense than the one at the higher wavenumber. However, the bands are most strongly enhanced in the Leu10 SERS spectrum, followed by Leu6, Leu7, and Leu4.

Based on these observations, and considering both the hybridization of P (sp^3^) and =O (sp^2^) atoms, as well as the down-shifting (Δ_ν_ = 9 and 4 cm^−1^), broadening (Δ_fwhm_ = ~6 cm^−1^), and relative intensity (measured against the intensity of the other bands in the spectrum) similar to and higher than the band at 1254 and 1265 cm^−1^ in the SERS spectra of Leu1 and Leu9 on AuNP, respectively, compared to this band in the FT-Raman spectra of the corresponding dipeptides (Figure 6), it is proposed that the free electron pair on the =O of Leu9 is in direct contact with the surface of the AuNP. This indicates that the P=O bond is inclined at an angle of 30° to the surface. In contrast, the same bond for Leu1 is inclined at a smaller angle to the surface of AuNP. This orientation facilitates interactions between the lone electron pair on oxygen in the P–OH fragment of the phosphate group and the surface of the AuNP. For the remaining dipeptides, only the P–OH fragment interacts with the AuNPs.

The SERS spectrum of Leu10 on AuNPs displays a band with the highest intensity at 1001 cm^−1^ [ν(PO)]. This is due to the perpendicular arrangement of the P–O bond with respect to the AuNP surface. In the series Leu10, Leu6, Leu4, and Leu7, the enhancement of the 1001 and 1026 cm^−1^ SERS signal [ρ_b_(POH)] decreases. However, the band at 1001 cm^−1^ is still more intense than the band at 1026 cm^−1^. These observations suggest that the P–O bond deviates from the surface normal of AuNP, causing a shift away from it. Although the 1026 cm^−1^ band is slightly more pronounced for Leu5, Leu1, and Leu9, both bands are stronger in the Leu5 SERS spectrum than in the Leu1 and Leu9 SERS spectra. These spectral features of Leu5 suggest that the slope of the P–O and O–H bonds relative to the AuNP surface is comparable, but in the case of Leu5, they are closer to it. In contrast, for the remaining dipeptides (Leu3, Leu2, and Leu8), the spectral feature at the higher wavenumber is much more intense than that at lower wavenumbers, indicating a less-perpendicular arrangement of the O–H bond with respect to the surface, compared with the P–O bond. The stronger intensity of these bands for Leu3 than for Leu2 and Leu8 suggests that these fragments are further away from the AuNP surface for Leu2 and Leu8.

The SERS spectra of the dipeptides studied also show bands related to amide modes. For example, in the SERS spectra of Leu2 and Leu9, Amide I is present as a weak band at 1688 and 1683 cm^−1^, respectively. Amide II is observed in the range of 1584–1566 cm^−1^ for Leu1, Leu3, Leu4, Leu5, Leu6, Leu8, and Leu9 (see Table 2 for detailed wavenumbers); however, it shows strong intensity only for Leu1 and Leu3. Bands at 1221, 1225, 1212, 1209, 1233, and 1209 cm^−1^, due to the Amide III mode, are observed in the SERS spectra of Leu10, Leu9, Leu8, Leu5, Leu3, and Leu1. Amide IV is enhanced for Leu2 (strong), Leu3, Leu6, Leu9, and Leu10, whereas Amide V is only present in the SERS spectrum of Leu10. From these observations, it can be concluded that Leu2 and Leu9 interact with the surface of AuNP through a lone electron pair on the oxygen of the amide bond. The oxygen of Leu9 appears to be closer to this surface than that of Leu2. The Leu10 dipeptide interacts with AuNPs through a lone electron pair on the amide nitrogen, resulting in a significant enhancement of the vibrations of the N_a_–H bond. The Leu8 and Leu3 dipeptides also make contact with the AuNP surface via a lone electron pair on the amide nitrogen, but with a different orientation that enhances the vibrations of N_a_–H and N_a_–C_a_. The remaining dipeptides interact weakly with AuNPs through their amide bond.

## 3. Materials and Methods

### 3.1. Peptide Synthesis

Phosphonate dipeptides of leucine (Leu) including [2-(2-amino-4-methylpentanamido)ethyl]phosphonic acid (Leu1), [2-(2-amino-4-methylpentanamido)propan-2-yl]phosphonic acid (Leu2), [2-(2-amino-4-methylpentanamido)butan-2-yl]phosphonic acid (Leu3), [1-(2-amino-4-methylpentanamido)propyl]phosphonic acid (Leu4), [(1S)-1-(2-amino-4-methylpentanamido)propyl]phosphonic acid, 1-(2-amino-4-methylpentanamido)-2-hydroxyethyl]phosphonic acid (Leu6), [1-(2-amino-4-methylpentanamido)-2,2-dimethylpropyl]phosphonic acid (Leu7), [(2-amino-4-methylpentanamido)(cyclopropyl)methyl]phosphonic acid (Leu8), 4-(2-amino-4-methylpentanamido)-4-phosphonobutanoic acid (Leu9), and [(2-amino-4-methylpentanamido)(adamantane)methyl]phosphonic acid (Leu10) were synthesized using a previously described procedure [99]. The peptides’ purity and chemical structures were confirmed using ^1^H, ^31^P, and ^13^C NMR spectra (Bruker Avance DRX 300 MHz spectrometer, Bruker Polska sp. z o.o., Poznań) and electrospray mass spectrometry (Finnigan 95).

### 3.2. Preparation of Silver Sol

AgNO_3_ and NaBH_4_ were purchased from Sigma-Aldrich Co. (Poznań, Poland) and used without further purification. Three batches of colloidal silver solution were prepared according to the standard procedure [95]. The procedure involved dissolving 8.5 mg AgNO_3_ in 50 mL of deionized water at 4 °C, and then adding it dropwise to 150 mL 1 mM NaBH_4_ while stirring vigorously. The resulting pale yellow solution was then stirred at 4 °C for approximately one hour. The produced AgNPs have a polydispersity index PDI < 0.25 (with DLS).

### 3.3. Preparation of Gold Sol

HAuCl_4_ and C_6_H_5_Na_3_O_7_ were purchased from Sigma-Aldrich Co. (Poznań, Poland) and used without further purification. A solution of colloidal gold was prepared three times according to the standard procedure [96]. First, 5 mg of HAuCl_4_ was dissolved in 50 mL of double-distilled water and brought to a boil. Then, 0.75 mL of a 1% sodium citrate solution was added. The yellow solution immediately turned dark blue and became dark red after boiling for 2 min. The produced AuNPs have a polydispersity index PDI < 0.2 (with DLS).

### 3.4. Ultraviolet-Visible Spectroscopy (UV-Vis)

UV-Vis spectra of an aqueous solution and sample/solution system were recorded using a Lambda 25 UV-Vis spectrometer after 120 min of mixing. The measurements were taken with a quartz cuvette that has a width of 1 cm. The colloid concentrations of 0.29 mM for Au and 0.35 mM for Ag were used.

### 3.5. Dynamic Light Scattering (DLS)

DLS analyses were performed on a Zetasizer Nano ZS analyzer (Malvern Instruments Ltd., Malvern, GB) equipped with a 4 mW He–Ne laser (633 nm), at a detection angle of 173°. Low-volume quartz cuvette ZEN2112 was used for size and particle concentration measurements, whereas cuvette DTS1070 was used for zeta potential measurements. The colloids concentration of 2 mg/mL was used.

### 3.6. X-ray Powder Diffraction (XRD)

The diffraction patterns were measured at room temperature using a Rigaku Ultima IV X-ray diffractometer (Rigaku Co., Tokyo, Japan) with CuK_α_ (λ = 1.542 Å) radiation at 40 kV and 40 mA. The diffraction patterns were collected in the range of 20–80° (2θ) (0.02°/step and 2 s per step).

### 3.7. X-ray Photoelectron (XPS)

XPS spectra were recorded for solid samples using a monochromatic Al K_α_ (E = 1486.7 eV) X-ray source on a Prevac photoelectron spectrometer equipped with a VG SCIENTA R3000 hemispherical analyzer. The NP dispersions were drop-cast and dried on silicon wafers. Prior to XPS measurements, the samples were discharged by electron gun neutralization. The binding energies were referenced to the C 1s nuclear level (285.0 eV).

### 3.8. Transmission Electron Microscopy with Energy-Dispersive X-ray Analysis (TEM-EDS)

The morphology and crystallite size of the samples were investigated using a Phillips CM20 TWIN transmission electron microscope operating at 200 kV working in a selected area electron diffraction (SAED) mode and equipped with an energy-dispersive X-ray (EDX) microanalyzer. To prepare samples for measurement, excess reagents were removed by centrifuging at 10,000 rpm for 10 min, and then one drop of the NPs was applied onto a carbon-coated copper grid. The grid was allowed to settle for five minutes before being dried.

### 3.9. Raman and Surface-Enhanced Raman (SERS)

Raman and SERS spectra were collected using an InVia Raman Spectrometer (Renishaw, Warsaw, Poland) equipped with an air-cooled charge-coupled device (CCD) detector and a Leica microscope (50× objective) and a HoloSpec f/1.8i spectrograph (Kaiser Optical Systems Inc., Ann Arbor, MI, USA) equipped with a liquid-nitrogen-cooled CCD detector (Princeton Instruments, Trenton, NJ, USA). The spectral resolution was set at 4 cm^−1^. The 785.0 nm line was used as an excitation source for AuNPs, while the 514.1 nm line was used for AgNPs. The laser power at the output was set to 20 mW.

Aqueous solutions of dipeptides were prepared by dissolving them in deionized water. The concentration of the samples was adjusted to 10^−4^ M before mixing with the sol. The freshly prepared sample solution was added to the sol to achieve a final sample concentration of approximately 10^−5^ M.

Colloids from three syntheses were used for the measurements. The SERS spectra of each dipeptide adsorbed on the surface of the nanoparticles were collected 9 times (from three drops of three batches of the given colloidal solution). There were no problems with the activity of the nanoparticles or the reproducibility of the results. The spectra were reproducible for the measurements performed up to 3 months. The SERS spectra were almost identical, with only minor differences (up to 5%) in some band intensities and positions (±2 cm^−1^).

### 3.10. Spectral Analysis and Graphics

Spectral analysis was performed using a freeware Spectragryph software for optical spectroscopy, version 1.2.16.1, from Dr. Friedrich Menges, Am Dummelsmoos 28, 87561 Oberstdorf, Germany.

The graphic (Figure 1) was made using ChemSketch (freeware), product version 2023.1.2, from www.acdlabs.com and a UCSF Chimera, product version 1.17.3 from the Regents of the University of California.

## 4. Discussion

The spectral analysis has revealed the following information:Leu1 interacts with the AgNP surface through an amide bond, which is arranged to allow direct contact between the lone pair on the nitrogen and the surface. Phosphonate and amine groups are located in close proximity to the surface. The dipeptide is adsorbed onto the AuNP surface via a phosphonate group accompanied by the N(H)–C_a_ fragment of the peptide bond.Leu2 is adsorbed on AgNPs via N(H)–C_a_(=O), while the PO_3_H^−^ and C–NH_2_ groups are in close proximity. On the AuNP surface, the phosphonate group of Leu2 forms an O–H bond that is almost perpendicular to the nearby AuNP surface, and there is also a C_a_=O bond in close proximity.The C_α_(NH_2_)C_a_(=O)N_a_(H) fragment of Leu3 is positioned on the AgNP surface, allowing for a weak interaction with the P=O group. In contrast, on the AuNP surface, the lone pair of electrons on the oxygen in the POH group interacts directly with AuNPs. This interaction occurs in a peptide orientation that allows for both an almost-vertical arrangement of the O–H bond and a C_a_–N(H) interaction with AgNPs.The lone pairs of electrons on the =O atoms of the amide bond and the P=O bond of the Leu4 peptide make direct contact with the AgNPs. In addition, the amide oxygen is in contact with the surface of AuNPs, but is located at a certain distance from the surface, similar to the vertically arranged O–H bond of the phosphonate group.Similar to the case of Leu4 on AgNPs, both free electron pairs of electrons on the O= atoms of Leu5 are in contact with the AgNP surface. However, the C_a_=O and P=O bonds are similarly tilted towards this surface. A lone pair of electrons on the oxygen of the P–O bond is also involved in the interactions between Leu5 and AgNPs. Leu5 is in contact with AuNPs via C_a_–N and POH fragments. Similarly, the P–O and O–H bonds are tilted towards the AuNP surface.Leu6 interacts with the AgNP surface via the POH fragment and amide bond. The bands of C–OH and NH_2_ groups indicate their proximity to AgNPs. In contrast, the interaction between AuNPs and Leu6 is caused by the vertical P–O bond at a certain distance from the AuNP surface, as well as the C_a_(=O)N_a_(H) fragment.The Leu7 fragment C_α_(N)C_a_(=O)N_a_CP(=O) interacts with the surface of AgNPs, while the vertical P–O bond mainly adsorbs onto AuNPs.The lone electron pair of the amide nitrogen is in contact with the AgNP surface, allowing the Na–H and C_a_=O bonds to be similarly tilted to this surface and the amino and phosphonate groups to be located near it. The C_a_NH fragment of Leu8 is also responsible for the interaction of this dipeptide with AuNPs. Regarding the surface, the interaction of Leu8 with the vertically arranged P–O bond is weak.The contact between Leu9 and the AgNP surface is mainly due to the R^2^ fragment, with support from the amide, P=O, and POH groups. In contrast, Leu9 adsorbs onto AuNPs mainly with P=O, POH (with a more or less vertical arrangement of the O-H bond), and Ca=O.The Leu10 fragment, specifically C_α_(N)C_a_(=O)N_a_, with weak involvement of the POH moiety, is accountable for the adsorption of Leu10 onto AgNPs. Additionally, the free electron pair on the oxygen of the vertically arranged P–O bond is primarily responsible for the peptide’s interactions with the surface of AuNP. The C_a_N_a_H fragment aids in this interaction.

## 5. Conclusions

This work describes positively charged gold and negatively charged silver nanoparticles prepared using an efficient and cost-effective synthesis route, which allows for obtaining highly pure, stable (non-aggregating without stabilizers) nanoparticles with a well-defined spherical shape, a desired diameter of 20 nm, and a monodispersion in an aqueous environment, as confirmed by spectroscopic studies. Nanoparticles with these parameters are important because of their antimicrobial and antiviral activity and in terms of their application as biomarker sensors and drug delivery platforms for precision treatment.

The surface of the nanoparticles was coated with leucine phosphonate dipeptides. Adsorption was monitored using the SERS technique, which resulted in reproducible spectra. Maintaining reproducibility is crucial for routine bioanalysis. The analysis aimed to determine the orientation of the dipeptides (coating layers) on the surface of the nanoparticles. It was observed that selecting the appropriate R^2^ side chain of dipeptides enables control over the arrangement of these dipeptides. This control allows for the proper formation of a layer that covers the nanoparticles while also interacting with the surrounding biological environment, such as cells, tissues, and biological fluids.

## Figures and Tables

**Figure 1 ijms-25-03720-f001:**
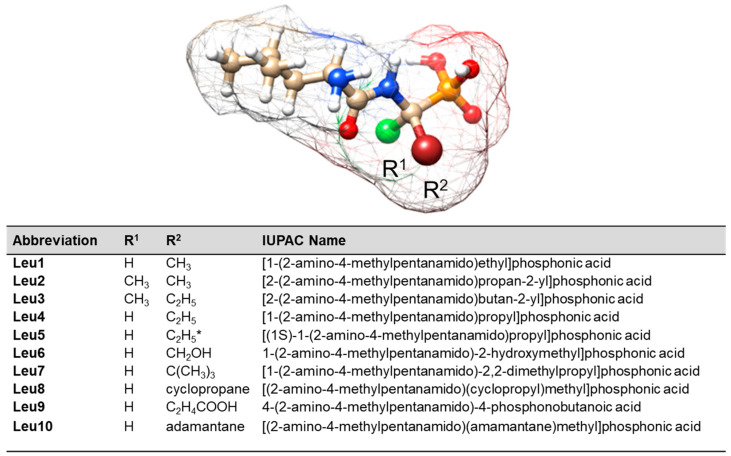
Structure of the investigated Leu1–Leu10 phosphonate dipeptides (structure generated using ChemSketch (freeware), product version 2023.1.2, from www.acdlabs.com and a UCSF Chimera, product version 1.17.3) (* denotes to *S* absolute configuration of a chiral molecular –NC(R^1^)(R^2^)P– entity; blue color denotes to nitrogen atoms, red—oxygen atoms, beige—carbon atoms, white hydrogen atoms, orange—phosphor atom, and green and dark red—substituents).

**Figure 2 ijms-25-03720-f002:**
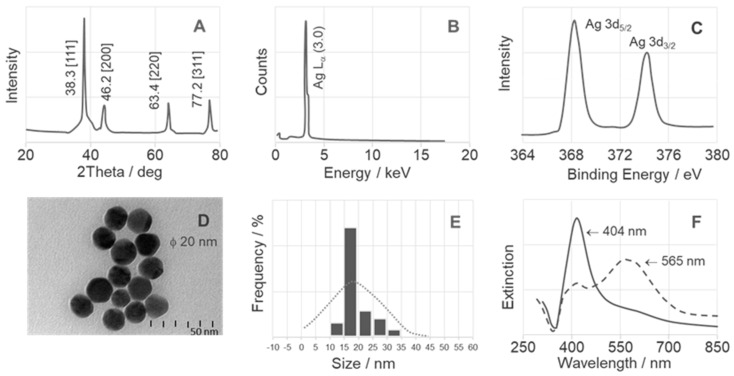
Surface analysis of AgNPs, with XRD–(**A**), EDS–(**B**), XPS–(**C**), TEM–(**D**), DLS–(**E**), and UV-VIS–(**F**).

**Figure 3 ijms-25-03720-f003:**
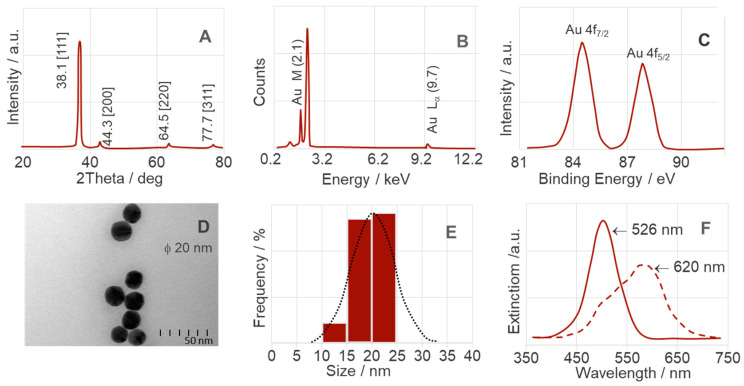
Surface analysis of AuNPs, with XRD–(**A**), EDS–(**B**), XPS–(**C**), TEM–(**D**), DLS–(**E**), and UV-VIS–(**F**).

**Figure 4 ijms-25-03720-f004:**
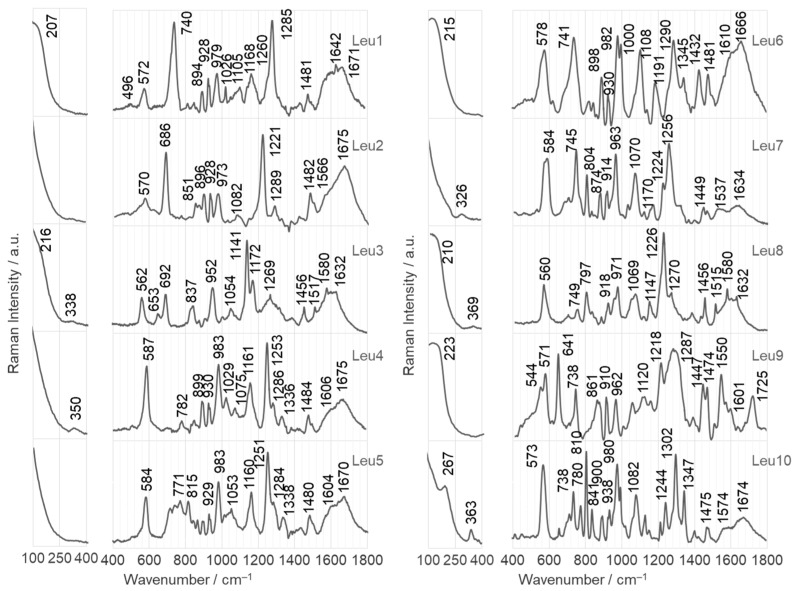
SERS spectra of Leu1–Leu10 phosphonate dipeptides immobilized on AgNPs surface.

**Figure 5 ijms-25-03720-f005:**
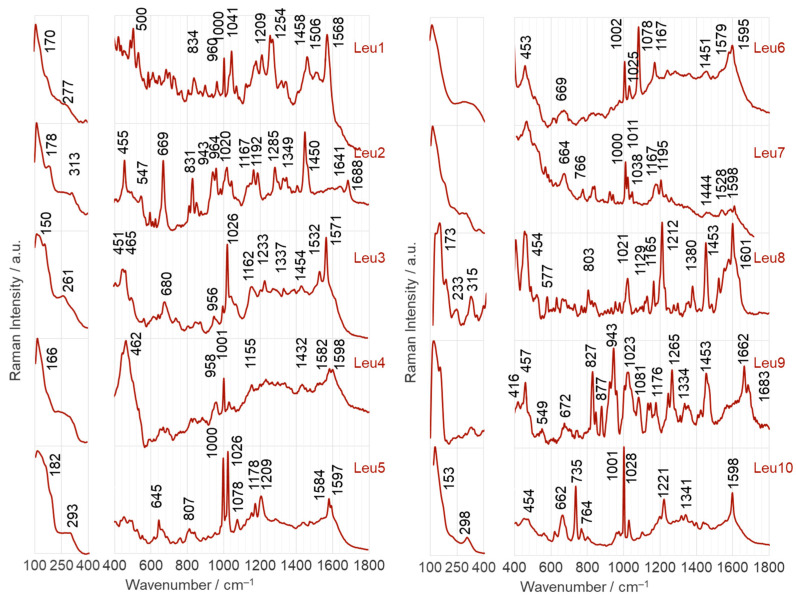
SERS spectra of Leu1–Leu10 phosphonate dipeptides immobilized on AgNPs surface.

**Figure 6 ijms-25-03720-f006:**
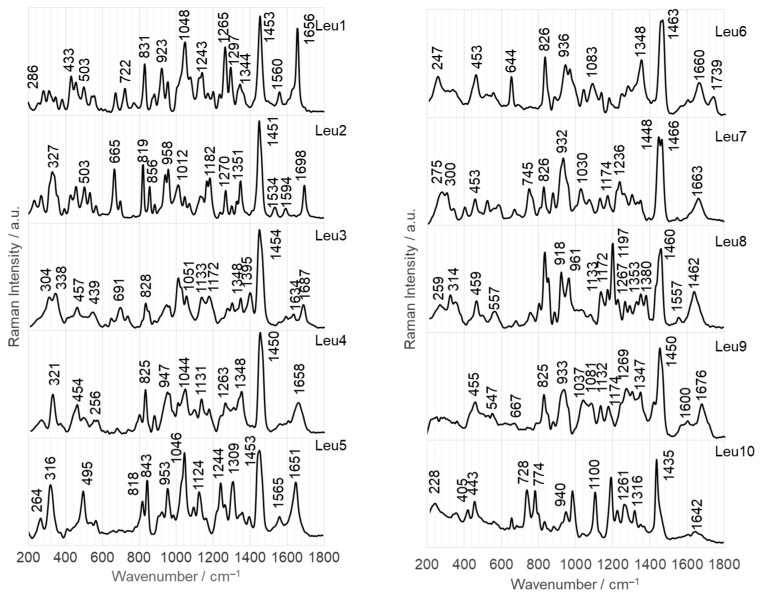
FT-Raman spectra of Leu1–Leu10 phosphonate dipeptides.

**Table 1 ijms-25-03720-t001:** Wavenumbers and band assignments for the SERS spectra of Leu1–Leu10 phosphonate dipeptides adsorbed onto AgNPs.

Assignment	Leu1	Leu2	Leu3	Leu4	Leu5	Leu6	Leu7	Leu8	Leu9	Leu10
Ag–molecule			216					210	223	
Ag–O			338	350			326	369		363
Ag–N	496									
C_a_(=O)N_a_CP									544	
Amide VI, ρ_oopb_(C_a_=O)	572	570	562	587	584	578	584	560	571	573
δ(N_a_C(P)C), δ_oop_(C_a_N_a_(H)C), ν(CP)									641	
AmideIV, ρ_ipb_(C_a_=O)		686	692							
AmideV, ρ_oopb_(NH)	740					741	745	747	738	738
ν(PO), ρ_r_/ρ_t_(CC(H,Y)C)					771					780
ν(PO)					815		804	797		810
ρ_r_(C_α_NH_2_), ρ_t_(C–P=O), ν(CN)		851							861	841
δ_oop_(C_α_NH_2_), ρ_t_(CPO_3_H^−^), ρ_t_(C_α_C_a_(=O)N_a_	894	896		899		898	874			900
ν(PO) + ρ_b_(POH)//**ν(C–COOH)**	928	928		930	929	930	914		**910**	938
δ_oop_(C_α_NH_2_)			952				963		962	
δ(CPO_3_H^−^), δ_oop_(C_α_NH_2_), ν(C_α_N)	979	973		983	983	982		971		980
ν(PO), ρ_b_(POH)						1000				1001
ρ_b_(POH), ν(C_α_N), ν(CC_α_)	1026		1054	1029	1053					
δ_oop_(C_α_NH_2_), ρ_r_(CCH_3_)_L_, ν(CC)_L_, ρ_t_/ρ_r_(CH_2_C_α_(H,N)C_a_		1082			1075		1070	1069		1082
ν(N_a_C), ρ_r_(CC(Y)C), δ(N_a_C(P)C), ν(CC), ρ_r_(CCH_3_)	1105					1108			1120	
ρ_r_(C_α_NH_2_), ν(CC_α_), ν(CN), ρ_r_ (C_α_(H,N)C_a_)			1141					1147		
ν(P=O)	1168		1172	1161	1160	1191				
AmideIII, ρ_ipb_(N_a_H) + ν(C_a_N_a_)		1221					1224	1226	1218	
ν(C_a_N_a_), ν(CN_a_)										1244
ν(P=O)	1260		1269	1253	1251	1262	1256	1270	1263	
ν(C_a_N_a_)//**ν(CO), ρ_b_(COH)**	1285	1289		1286	1284	**1290**			1287	1302
ρ_r_(CC(H,C)C_β_(H_2_)C_α_(H,N)C_a_)				1336	1338	1345				1347
δ/ρ_r_(CCH_3_)//**ρ_w_/ρ_t_(CCH_2_O), ρ_b_(COH)**						**1432**		1456	**1447**	
δ/ρ_r_(CH_3_)	1481	1482		1484	1480	1481			1474	1475
AmideII, ρ_ipb_(N_a_H) + ν(C_a_N_a_)	1580	1566	1580					1580	1550	1574
δ(NH_2_), ρ_b_(C-NH_2_)	1642		1632	1606	1604	1610	1634	1632	1601	
AmideI, ν(C_a_=O)	1671	1675		1675	1670	1666				1674
**ν(COOH)**									**1725**	

“_a_” denotes the amide atom; C_α_—α-carbon; “ρ_ipb_”—in-plane bending, “ρ_b_”—bending, “ρ_t_”—twisting, “ρ_r_”—rocking, and “δ”—deformation vibrations; (…)_L—_a fragment of *N*-terminal L-leucine; Y—the H, C, N, or P atoms; **bold format**—bands assignment for compounds consisting of –COOH or –OH group.

**Table 2 ijms-25-03720-t002:** Wavenumbers and band assignments for the SERS spectra of Leu1–Leu10 phosphonate dipeptides adsorbed onto AuNPs.

Assignment	Leu1	Leu2	Leu3	Leu4	Leu5	Leu6	Leu7	Leu8	Leu9	Leu10
Ag–molecule	170	178	150	166	182			173		153
Ag–O	277	313	261		293			233,315	320	298
Ag–N, C_a_(=O)N_a_CP	500	455	458	462		453	451	405,454	457	454
Amide VI, ρ_oopb_(C_a_=O)		547							549	
δ(N_a_C(P)C), δ_oop_(C_a_N_a_(H)C), ν(CP)					645					
AmideIV, ρ_ipb_(C_a_=O)		669	680			669	664		672	662
AmideV, ρ_oopb_(NH)							766			735,764
ρ_r_(C_α_NH_2_), ρ_t_(C–P=O), ν(CN)	834	831							827	
δ_oop_(C_α_NH_2_), ρ_t_(CPO_3_H^−^), ρ_t_(C_α_C_a_(=O)N_a_									877	
δ_oop_(C_α_NH_2_)//**ν(C–COOH)**		943	956	958					943	
δ(CPO_3_H^−^), δ_oop_(C_α_NH_2_), ν(C_α_N)	960	964								
ν(PO), ρ_b_(POH)	1000	1001	1000	1001	1000	1002	1000	1001	1002	1001
ρ_b_(POH)	1041	1020	1026	1030	1026	1025	1011	1021	1023	1028
δ_oop_(C_α_NH_2_), ρ_r_(CCH_3_)_L_, ν(CC)_L_, ρ_t_/ρ_r_(CH_2_C_α_(H,N)C_a_					1078	1078			1081	
ρ_r_(C_α_NH_2_), ν(CC_α_), ν(CN), ρ_r_(C_α_(H,N)C_a_), ν(P=O	1162	1167	1162	1155		1167	1167	1165	1176	
AmideIII, ρ_ipb_(N_a_H) + ν(C_a_N_a_)	1209	1192	1233		1209		1195	1212		1221
ν(P=O)	1254								1265	
ν(C_a_N_a_)//**ν(CO), ρ_b_(COH)**		1285								
ρ_r_(CC(H,C)C_β_(H_2_)C_α_(H,N)C_a_)		1349	1337						1334	1341
δ/ρ_r_(CCH_3_)//**ρ_w_/ρ_t_(CCH_2_O), ρ_b_(COH)**	1458	1450	1454	1432		**1451**	1444	1453	**1453**	
δ/ρ_r_(CH_3_)_L_, δ(CCH_3_)_L_, δ(CC(H_2_)C_α_), ρ_r_(C_α_(H,N)C_a_)	1506		1532				1528	1528		
AmideII, ρ_ipb_(N_a_H) + ν(C_a_N_a_)	1568	1588	1571	1582	1584	1579		1581		
δ(NH_2_), ρ_b_(C-NH_2_)				1598	1597	1595	1598	1601		1598
AmideI, ν(C_a_=O)		1641							1662	
**ν(COOH)**									**1683**	

“_a_” denotes the amide atom; C_α_—α-carbon; “ρ_ipb_”—in-plane bending, “ρ_b_”—bending, “ρ_t_”—twisting, “ρ_r_”—rocking, and “δ”—deformation vibrations; (…)_L—_a fragment of *N*-terminal L-leucine; Y—the H, C, N, or P atoms; **bold format**—bands assignment for compounds consisting of –COOH or –OH group.

## Data Availability

Data are available upon request (proniewi@agh.edu.pl).

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
