# Peer review of "Gold and Silver Nanoparticles as Biosensors: Characterization of Surface and Changes in the Adsorption of Leucine Dipeptide under the Influence of Substituent Changes"

_ijms, 2024, doi:10.3390/ijms25073720_

Round 1
Reviewer 1 Report
Comments and Suggestions for Authors
The article presents a comprehensive study on the synthesis and characterization of phosphonate dipeptides and their interactions with colloidal silver and gold nanoparticles using various spectroscopic and microscopy techniques, shedding light on their potential applications in nanotechnology and biomedicine. The following comments and questions can help improve the article.
1- The number of keywords is too high.
2- I suggest sections two and three to be swapped.
3- What specific rationale guided the selection of L-leucine phosphonate dipeptides for surface functionalization?
4- Did the study employ techniques such as dynamic light scattering (DLS) to complement the information obtained from TEM and XRD?
5- Surface functionalization of nanoparticles can sometimes lead to challenges such as non-specific binding or steric hindrance. Did the study encounter any difficulties related to the surface modification process, and if so, how were these challenges addressed?
6- Improve your introduction by referring to other users of gold and silver nanoparticles in sensors (DOI: 10.1016/j.jsamd.2024.100696 and DOI: 10.1088/1361-6528/acc6d7).
7- Have you investigated the reproducibility of SERS measurements, especially in terms of peak intensity and position?
8- Consider performing time-resolved SERS experiments to monitor dynamic processes, such as peptide adsorption/desorption kinetics on nanoparticle surfaces.
Author Response
Comment 1. The number of keywords is too high.
Author’s replay: As suggested by the Reviewer, the list of keywords has been shortened (page 1, highlighted).
Comment 2. I suggest sections two and three to be swapped.
Author’s replay: As suggested by the Reviewer, sections 2. and 3. have been swapped (pages: 4-6, Section 2. Materials and Methods; pages: 6-14, Section 3. Results).
Comment 3. What specific rationale guided the selection of L-leucine phosphonate dipeptides for surface functionalization?
Author’s replay: On pages 3-4, it is explained why phosphonate dipeptides of leucine were used to functionalize the surface of silver and gold nanoparticles. These dipeptides combine unique properties resulting from the presence of a phosphonamine group and leucine itself (highlighted).
Comment 4. Did the study employ techniques such as dynamic light scattering (DLS) to complement the information obtained from TEM and XRD?
Author’s replay: Yes, dynamic light scattering (DLS) analysis was used to complement the information obtained from TEM and XRD. The manuscript has been updated accordingly (see section 2.5. Dynamic Light Scattering (DLS) on page 5 (highlighted) and section 3.1. Surface analysis of NPs on pages: 6-7).
Comment 5. Surface functionalization of nanoparticles can sometimes lead to challenges such as non-specific binding or steric hindrance. Did the study encounter any difficulties related to the surface modification process, and if so, how were these challenges addressed?
Author’s replay: Our research involves the development of methods for obtaining nanoparticles and functionalizing their surfaces with biomolecules, including amino acids, dipeptides, peptides, and proteins. We do not use nanoparticle stabilizers or surfactants. We change the incubation time before measuring. In this paper’s studies, we encountered no problems with functionalization or reproducibility.
Comment 6. Improve your introduction by referring to other users of gold and silver nanoparticles in sensors (DOI: 10.1016/j.jsamd.2024.100696 and DOI: 10.1088/1361-6528/acc6d7).
Author’s replay: As suggested by the Reviewer, studies with the following DOIs have been included (Refs. 41 and 42, pages: 2 and 17, highlighted).
Comment 7. Have you investigated the reproducibility of SERS measurements, especially in terms of peak intensity and position?
Author’s replay: The reproducibility of results is crucial in the case of SERS. Therefore, this study has considered the reproducibility of the results. Section 2.9. Raman and surface-enhanced Raman (SERS) has been rewritten to clarify this (pages 6, highlighted).
Comment 8. Consider performing time-resolved SERS experiments to monitor dynamic processes, such as peptide adsorption/desorption kinetics on nanoparticle surfaces.
Author’s replay: Our experience has shown that some relatively small biomolecules, such as Gly-Gly dipeptide (Appl. Spectrosc., 58, 570-580 (2004)) and alpha−aminophenylphosphinic acid of pyridine [(butylamino)(pyridin-2-yl)methyl]phenyl-phosphinic acid (Appl. Surf. Sci., 563 (2021) 150341), etc.) exhibit time-dependent adsorption on a metal surface. For this reason, we typically examine biomolecules for this behavior. In the case of the compounds studied, no time-dependence was observed.

Reviewer 2 Report
Comments and Suggestions for Authors
The current manuscript is an interesting experimental study on the development of silver and gold nanoparticles as biosensors. It appears to be overall well-done, with many relevant assays having been performed. Hence, only some alterations are necessary before acceptance for publication:
- The iThenticate report shows a high degree of similarity (25%) with existing sources, hence the text should be rewritten in the identified sections;
- The keywords are not adequately defined, they are too many, and each almost represents a description instead of a keyword, this should be corrected (less words, more summarization);
- In the sentence “A nanoparticle (NP) is “a natural, incidental, or manufactured material containing particles, in an unbound state or as an aggregate or agglomerate and in which 50% or more of the particles in the size distribution of the number have one or more external dimensions in the size range 1 nm–100 nm” [1] the expression between quotation marks should be rewritten and replaced by the authors own words; full sentences from references should be discouraged;
- Figure 1 caption should mention the source of the molecule: did it derive from another publication, or was it made with a specific software? More detail should be provided;
- Materials and Methods subsections should be adequately numbered;
- The used methods should be better described, in more detail; for example for particle size distribution cuvette type and composition and sample dilution should be provided;
- The conclusion section is too long; this section should actually be the “Discussion” section, and then a much more summarized “Conclusion” section should be added at the end of the manuscript;
- In the discussion section, this study’s results should be compared to other previous studies using similar methods; the appropriate references should also be added;
- Nanoparticle PDI (polydispersity index) should be reported;
- Safety issues regarding silver and gold nanoparticle administration should be discussed;
- An abbreviation list should be added.
Author Response
Referee #2
Comment 1. The iThenticate report shows a high degree of similarity (25%) with existing sources, hence the text should be rewritten in the identified sections.
Author’s replay: As recommended by the reviewer, the work has been reviewed and checked for plagiarism using Duplichecker.com. The analysis revealed that 16.3% of the text (1629 words) is similar to other texts. Out of these, 1035 words are related to the references, 29 words correspond to formulas with acknowledgments and affiliation, and there are also similarities in the Materials and Methods section. The cited papers cannot be altered in terms of their titles, author names, or scientific apparatus used in the research. Some similarities in the introduction are inevitable due to the author's style or the brief description by own author words of other researchers' studies, which may include characteristic doses, names of compounds and spectroscopic methods, or symptoms.
Comment 2. The keywords are not adequately defined, they are too many, and each almost
represents a description instead of a keyword, this should be corrected (less words, more summarization);
Author’s replay: As suggested by the Reviewer, the keyword list has been shortened and corrected (page 1, highlighted).
Comment 3. In the sentence “A nanoparticle (NP) is “a natural, incidental, or manufactured material containing particles, in an unbound state or as an aggregate or agglomerate and in which 50% or more of the particles in the size distribution of the number have one or more external dimensions in the size range 1 nm–100 nm” [1] the expression between quotation marks should be rewritten and replaced by the authors own words; full sentences from references should be discouraged.
Author’s replay: As suggested by the Reviewer, the cited sentence has been rewritten (page 1, highlighted).
Comment 4. Figure 1 caption should mention the source of the molecule: did it derive from another publication, or was it made with a specific software? More detail should be provided.
Author’s replay: The author has addressed the Reviewer's comment regarding the need for more detail in the caption of Figure 1. The source of the molecule has been mentioned in the caption of Figure 1. The author has also added the necessary information to section 2.10. Spectral analysis and graphics (page 6, highlighted).
Comment 5. Materials and Methods subsections should be adequately numbered.
Author’s replay: The Materials and Methods subsections have been adequately numbered, as suggested by the Reviewer (pages 4-6).
Comment 6. - The used methods should be better described, in more detail; for example for particle size distribution cuvette type and composition and sample dilution should be provided.
Author’s replay: Appropriate information has been added to the research methods section, as suggested by the Reviewer (pages 4-6).
Comment 7. The conclusion section is too long; this section should actually be the “Discussion” section, and then a much more summarized “Conclusion” section should be added at the end of the manuscript.
Author’s replay: The author has responded to this suggestion by adding the Discussion section. The comparison of the adsorption of the tested compounds on the surfaces of AuNPs and AgNPs has been relocated to pages 14-15.
Comment 8. In the discussion section, this study’s results should be compared to other previous studies using similar methods; the appropriate references should also be added.
Author’s replay: As recommended by the Reviewer, previous studies have been mentioned and referenced (page 8, highlighted).
Comment 9. Nanoparticle PDI (polydispersity index) should be reported.
Author’s replay: As suggested by the Reviewer, the PDI has been included (page 5, Sections 2.2. and 2.3.).
Comment 9. Safety issues regarding silver and gold nanoparticle administration should be discussed.
Author’s replay: Information on the safety administration of AgNPs and AuNPs has been added, as suggested by the reviewer (see page 2, highlighted).
Comment 10. An abbreviation list should be added.
Author’s replay: An abbreviation list has been added as suggested by the reviewer (pages 15-16, highlighted).
